# Long-Term Recording of Reticulo-Rumen Myoelectrical Activity in Sheep by a Telemetry Method

**DOI:** 10.3390/ani11041052

**Published:** 2021-04-08

**Authors:** Małgorzata Wierzbicka, Małgorzata Domino, Romuald Zabielski, Zdzisław Gajewski

**Affiliations:** Department of Large Animal Diseases and Clinic, Institute of Veterinary Medicine, Warsaw University of Life Sciences, 02-797 Warsaw, Poland; romuald_zabielski@sggw.edu.pl (R.Z.); zdzislaw_gajewski@sggw.edu.pl (Z.G.)

**Keywords:** electromyography, rumen, reticulum, rumination cycle, ewes

## Abstract

**Simple Summary:**

The motility of the smooth muscle of the rumen and the reticulum is tightly synchronized since they are responsible for degrading the ingested fibrous diet. It involves mixing contractions of partially digested material (cycle A), cyclic eructation of fermented gases (cycle B), and contractions associated with regurgitation and rumination (cycle C). Disorders of cycles A, B, and C in reticulo-rumen contraction occurred in cases of many metabolic diseases. Since the smooth muscle electromyography (EMG) is closely associated with the motility pattern, our study aimed to implement the long-term EMG telemetry recording method to analyze the basic reticulo-rumen motility pattern in conscious unrestrained sheep. The study was conducted on nine ewes chronically fitted with bipolar electrodes in the rumen and reticulum. Cycles A and B occurred constantly in a similar share, whereas cycles C appeared every 30 min and lasted for the most part 15–30 min. EMG signal analysis showed more differences at the level of single bursts than on burst bundles. Recording reticulo-rumen EMG signal with a long-term telemetry approach offers a reliable tool to assess the forestomach motility in conscious, unrestrained sheep, allowing for easy recognition of reticulo-rumen cycles.

**Abstract:**

The reticulum and rumen are considered a single functional unit (the reticulo-rumen) with regards to myoelectrical and contractile activities. The specialized contractions of the reticulo-rumen provide constant mixing of partially digested material (cycle A), its flow into the omasum during eructation (cycle B), and regurgitation-rumination (cycle C). This study aimed to investigate the feasibility of electromyography (EMG) registered by a long-term telemetry method for assessment of the basic reticulo-rumen myoelectrical activity in sheep, to develop the effective recognition of the reticulo-rumen cycles at rest with no food stimulation, and to investigate the relationship between cycles A, B, and C in such basic conditions. The experiment was carried out on nine ewes. Myoelectric activity of the rumen, reticulum, and abomasum was recorded by the combination of three silver bipolar electrodes and a 3-channel transmitter implant. The myoelectrical activity registered successfully in the reticulum and rumen was determined as three characteristic patterns of cycles A, B, and C. The percentage of each type of cycle changed at different intervals from equally cycles A (43–50%) and B (50–56%), occurring when cycle C was not observed to the domination of cycle C (57–73%) with a decrease of cycles A (6–14%) and B (20–28%). The long-term EMG telemetry registration is feasible in the assessment of the reticulo-rumen myoelectrical activity in sheep.

## 1. Introduction

In ruminants, the myoelectric activity of the smooth muscle of the rumen and the reticulum is characterized by contractility that allows intake, degradation, and transport of various fibrous components and their metabolic products. This is followed by evacuation of fermented gases. The reticulum and rumen are considered a single functional unit when it comes to myoelectrical and contractile activities. This is because their contractions and their control are primarily vagal-dependent, they are initiated in the reticulum and then their myoelectrical signal is transferred to the rumen [1,2,3]. This activity, known as the reticulo-rumen cycles pattern, allows mixing and degradation of ingested plant material, eructation of fermented gases, separation, and transport of plant particles. The latter is done either backwards for rumination purposes or continuously for further digestive processes in the omasum, abomasum, and intestines [3,4]. Due to the alternating contractions of the rumen and the reticulum, the insoluble heavier, crumbled nutrients (e.g., cereal grains) fall to the bottom of the ventral rumen sac, while the lighter, finely divided components form a spongy weave of fibers on the surface of the rumen, over which fermented gases accumulate [5]. In the reticulo-rumen cycles, regular contraction sequences within individual parts of the stomach chambers are organized in three schemes [5,6]. The specialized contractions of the reticulo-rumen provide constant mixing of partially digested material (cycle A) and its flow into the omasum as well as cyclic eructation (cycle B) and regurgitation-rumination (cycle C). Braun et al. [7,8] used the ultrasonographic examination to describe A and B cycles, differentiating between primary and secondary rumen contractions. Additionally, Plaza et al. [4] used the terms primary and secondary contractions to describe mixing and eructation in the reticulo-rumen using electromyography (EMG). However, the authors often reported additional contractions of the reticulum during rumination. Until now, the involvement of the rumen and of the reticulum in contraction cycles as well as the correlation between mixing/eructation and rumination contractions was unclear. Therefore, this preliminary study aimed to record EMG signals in the reticulum and the rumen of conscious unrestrained sheep with telemetry method. This investigation will enable further long-term study on the changes of the reticulo-rumen cycles in response to different diets and/or pharmaceutical treatments.

The main advantage of the present approach is the long-term recording of EMG signals, which allows recognizing cycles A, B, and C at rest. The forestomach motility in ruminants can be assessed physiologically using direct and indirect methods. Indirect methods, such as ultrasonography [7,8] and bolus wireless sensors with an accelerometer [9,10], have some limitations. Bolus sensors have numerous artifacts whereas ultrasonography approaches do not provide easy access to all components. This is turn skews the actual contraction of the examined part of the gastrointestinal tract. On the other hand, the direct methods, such as the force transducer method [11,12] or electromyography approach [4,13], require surgery. However, this can record continuous measurements over a long time and allow to obtain a signal with the least number of artifacts. The EMG method is characterized by a high correlation with the motor activity of the examined structures and high repeatability of results [14]. The EMG method, when the signal is collected directly from the muscle layer of the forestomach via electrodes, has been recognized as effective and reliable in sheep [15,16]. In later studies, the usefulness of EMG as a method that reflects the electrical activities of myocytes in the muscular layer of the gastrointestinal wall was confirmed as the most reliable source describing motor activity [4,13]. The most important limitation concerns signal transmission. Kania et al. [17], Sarna et al. [18], and Plaza et al. [4] used cable signal transmission, which caused long-term immobilization of the animals and limited the duration of registration from 6 [5,18] to 12 h [4]. In contrast, considering all available methods, ultrasonography allowed us to exam the few minutes of forestomach activity [7,8], whereas, the telemetry bolus-type sensor enable a few hours of observation [9,10,12]. Therefore, this study presented EMG registration using long-term telemetry method of signal transmission. The long-term telemetry method allows to significantly extend the time of registration from several days to around 2 months [19,20,21]. The long-term telemetry method enabled EMG measurements which maintained animal welfare and reduced the animal’s stress associated with immobilization during registration [13,22]. However, it should be kept in mind that the advantage of telemetry may not even compensate for the surgery necessary to implant the electrodes. The long-term telemetry method has been successfully used to study the myoelectric activity of the gastrointestinal tract in pigs [13,23] and the reproductive tract in pigs [22,24,25,26]. Therefore, the implementation of the long-term telemetry method to evaluate the reticulo-rumen myoelectrical activity in sheep was proposed in this paper.

There is a need for introducing an accurate registration and transmission method that allows proper identification and description of cycles A, B, and C of the rumination process, due to the importance of forestomach diseases. Motility disorders of the forestomach which accompany many diseases of bacterial and metabolic etiology cause severe pain symptoms and are an important clinical problem in ruminants [27]. Gas accumulation in the rumen results in free gas bloat observed in hypocalcemia [28] or grain overload [29] and are consequences of inhibition of cycle B, which are vital for eructation. On the other hand, improper mixing of the digest in the rumen (cycle A) and rumination (cycle C) could result in the rumen overload and impaction on the rumen and omasum [30]. In many metabolic diseases, such as ruminal acidosis [31], hypocalcemia [11], and numerous diseases causing symptoms of severe pain [32], disturbances of the frequency and amplitude of contractions of the reticulo-rumen were reported. Therefore, scientific research on the myoelectrical activity of the gastrointestinal tract is needed to prevent clinical cases of gastrointestinal motility disorders, mainly of the forestomach in ruminants [27,33,34]. However, the EMG evaluation of the reticulo-rumen motility cannot be considered as a practical solution in clinics or agriculture.

The primary objective of this study is to investigate the feasibility of electromyography registered by a telemetry method for the assessment of the reticulo-rumen myoelectrical activity in sheep. Ancillary goals were to develop the effective recognition of the reticulo-rumen cycles at rest with no food stimulation, and investigate the relationship between cycles A, B, and C in such basic conditions. To achieve this goal, the reticulo-rumen contractile pattern at rest was characterized in detail in nine ewes based on the amplitude, root mean square (RMS), duration, and frequency of myoelectrical activities.

## 2. Materials and Methods

### 2.1. Animals

The experimental protocol was approved by the II Local Ethical Committee on Animal Testing in Warsaw (Permit Number: 14/2013, from 27 March 2013) on behalf of the National Ethical Committees on Animal Testing. The experiment was carried out on nine mature crossbred ewes (*n* = 9), each weighing 35–40 kg body weight. The ewes were adapted to the animal facilities for 7 days before the experiment. During the whole experiment, the ewes were kept in individual cages and had unlimited tap water and hay. During the entire study, the ewes were clinically healthy. Myoelectric activity of the rumen, reticulum, and abomasum was recorded by the combination of three silver bipolar electrodes [13,18] connected to a 3-channel transmitter implant (TL10M3-D70-EEE, DSI, St. Paul, MN, USA). In brief, the bipolar needle electrodes were made from a silver wire (0.4 mm, Ag 99.98% purity, Mennica Polska, Warsaw, Poland) and connected to a stranded type steel wire immersed in Teflon (A-M Systems, Inc., Sequim, WA, USA). The electrodes were then fixed between two silicone foil pieces (Σ Sigma medical, No. 2079, Nanterre, France) with silicone glue (Silastic^®^ No. 891, Dow Corning, Midland, MI, USA). A set of three electrodes was connected to the 3-channel transmitter implant. Before the surgery, the electrodes and the implant were sterilized in a 2% solution of glutaraldehyde (Aldesan E, Septoma, Warsaw, Poland). They were then washed with aseptic physiological saline (0.9% NaCl, Polfa S.A., Lublin, Poland).

Implantation of electrodes and transmitter was performed under general isoflurane anesthesia [35] in animals premedicated with midazolam (Midanium, 0.25 mg/kg b.wt., i.m., Polfa S.A., Lublin, Poland). Following midline laparotomy, the reticulum, abomasum, and rumen were extracted from the abdominal cavity, and electrodes were sutured, on the central section of the cranial part of the reticulum, on ventral sac of rumen, and abomasum (4 cm to pylorus), respectively. The electrode on the abomasum was used to follow the passage of ingesta from the reticulo-rumen to the abomasum. The electrode on the abomasum did not affect the myoelectrical activity of the reticulo-rumen, nor the classification of signals as mixing, eructation, or rumination patterns. In the next step, the telemetry transmitter TL10M3-D70-EEE (DSI, St. Paul, MN, USA) was fixed in the pocked between the abdominal muscles on the right flank, a ground electrode was sutured to the abdominal muscles near the transmitter, and the laparotomy was routinely closed. In the recovery period, ewes were treated with tolfenamic acid (Tolfedine, 2 mg/kg b.wt., i.m., Vetoquinol, Lure, France) and procaine penicillin with dihydrostreptomycin (Pen-strept, 10 mg/kg b.wt., i.m., Scanvet, Dublin, Ireland) for 5 days. The healing was fast and without complications.

### 2.2. Electromyography Recording and Analysis

Control EMG recordings started the next day after the surgery, however, the EMG signal free of injury potentials was observed just a few days after surgery, in general, from the 3rd to 5th day after the surgery. Regular EMG recordings lasted usually for 5–6 weeks. After that time the increasing number of artifacts made the EMG recordings useless for analysis. Before each EMG recording session, the access to feed was limited for 12 h before the beginning of recording with free access to water. Registration started at 7 a.m., and it was conducted for 3 h. To spare the battery, after each recording session, the telemetry transmitter implant was turned off. Recorded with 100 Hz sampling frequency EMG signal was digitized and modulated into radiowave and received by a receiver (RMC-1, DSI, New Brighton, MN, USA) located nearby the animal’s cage. The radiowave signal was then demodulated, amplified (Matrix, DSI, St. Paul, MN, USA), and archived (Neuroscore, DSI, St. Paul, MN, USA) for further off-line analysis. The EMG signals were digitally filtered with a band-pass filter (5–50 Hz) [36,37,38]. The myoelectrical activity of the rumen and reticulum was defined as a series of spike potentials with an amplitude exceeding 5 μV and duration longer than 0.5 s [24]. Spiking activities were organized into bursts and bundles forming A, B, and C cycles. Bursts were defined as series of spiking activities with amplitude exceeding 5 μV. In bursts, spike activities occurred one after the another with pause between spikes shorter than 0.5 s. The next burst began when the pause between subsequent spikes was longer than 0.5 s [4]. Bundles were defined as series of bursts with amplitude exceeding 5 μV and pauses between bursts longer than 5 s. The myoelectrical activity of the rumen and the reticulum was analyzed using the following parameters: burst mean amplitude (mV), burst mean RMS (root mean square) (mV), burst duration (s), bundle mean amplitude (mV), bundle mean RMS (mV), bundle duration (s), and a number of bursts forming a bundle. Parameters characterizing individual A, B, and C cycles were described in 12·15-min intervals (T1–T12) for each 3 h recording session.

### 2.3. Electromyography Recognition of A, B, and C Cycles

The classification of signals as mixing, eructation, and rumination patterns was proposed basing on Plaza et al. [4] and Braun et al. [7,8] works. Braun et al. [7,8] described A and B cycles, as primary and secondary rumen contraction. Each primary contraction cycle (cycle A—mixing) starts with biphasic reticular contraction, which is followed by contraction of the anterior blind sac of the rumen and then contraction of the dorsal sac of the rumen in a cranio-caudal direction. Each secondary contraction cycle (cycle B—eructation of gas) does not involve the reticulum and ruminal atrium and consists of contraction of the dorsal sac and then the ventral sac of the rumen. Therefore, using electrode arrangement presented in this study, cycle A was identified as the reticulum monophasic or biphasic contraction that occurred before the single rumen contraction, whereas cycle B was identified as the reticulum monophasic or biphasic contraction that occurs before the double rumen contraction.

Additionally, Plaza et al. [4] used the terms primary and secondary contractions to describe mixing and eructation in the reticulo-rumen using electromyography. However, the authors proved during rumination, the reticular biphasic contraction became triphasic. It was the first borderline criterion for the identification of the C cycle. Braun et al. [7] confirmed Plaza’s et al. [4] findings that a triphasic contraction of the reticulum occurs when a biphasic contraction is immediately preceded by a so-called rejection contraction, which function is to transport the cud into the esophagus, and it represents the rumination. It was the second borderline criterion for the identification of the C cycle in our work. Therefore, using electrode arrangement presented in this study, cycle C was identified as the reticulum triple contraction that occurs before the rumen contraction, whether the contraction of the rumen was single or double.

### 2.4. Data Analysis

All statistical evaluation was performed by GraphPad Prism6 software (Graph-Pad Software, San Diego, CA, USA), and the level of statistical significance was set to *p* < 0.05. The non-parametric Kolmogorov–Smirnov test, one-way analysis of variance (ANOVA) supported by the Holm–Sidak multiple comparison test or Kruskal–Wallis rank ANOVA supported by the Dunn multiple comparison test were used to compare the tested features between intervals (T1–T12) and to compare more than two features at individual intervals and Student’s *t*-test with Welch correction or Mann Whitney’s test to compare two features at individual intervals. The results in Table 1 were reported as mean ± SD. Additionally, the variability across animals and variability over the 5–6 week period of observations were calculated using the coefficient of variation (CV, %). First, the CVs were calculated across animals for each day, and mean values are presented in Table 2. Then, the CVs were calculated throughout observations for each animal, and mean values are presented in Table 3.

## 3. Results

The myoelectrical activity registered in the reticulum and rumen was determined as a three-cycle pattern using burst and bundle parameters. Cycle A (mixing) was defined as single or double bursts forming bundles in the reticulum (limit values: A > 5 μV; T > 0.5 s), before occurrence of single bundle in the ventral rumen sac (limit values: A > 5 μV; T > 2 s) (Figure 1). Cycle B (eructation) was defined as single or double bursts forming bundles in the reticulum (limit values: A > 5 μV; T > 0.5 s), then double bursts forming bundles in the ventral rumen sac (limit values: A > 5 μV; T > 2 s) (Figure 2). Cycle C (regurgitation-rumination) was determined as triple bursts forming bundlez in the reticulum (limit values: A > 5 μV; T > 0.5 s), followed by single or double bursts forming bundles in the ventral rumen sac (limit values: A > 5 μV; T > 2 s) (Figure 3).

During the preprandial period, the number of ruminal cycles remained constant 12 ± 0.2 to 16 ± 0.1 in 15 min intervals, which means that they appeared with frequency 0.013–0.018 Hz. Cycles A and B were observed in all 12 intervals, whereas cycle C was observed only in T3–4, T6–8, T11–12. We found that the percentage of each type of cycle changed at different intervals during the rest period. In the rest period, cycles A and B occurred equally with the percentages of 43.75–50.00% and 50.00–56.25% respectively, when cycle C was not observed. Cycle C appeared every 30 min for at least two intervals and in that time were dominating cycles (57.14–73.33% of all cycles recorded). At this time only intermittent cycles A (6.67–14.29%) and B (20.00–28.57%) were observed. In the summary of observations, the repeated occurrence of the rumination phase during 30 min intervals was repeatable in all animals from the start point of synchronization (Figure 4).

The EMG parameters of each cycle were characterized in detail in the reticulum and the ventral sac of the rumen (Table 1), and the variability across animals and variability throughout observations are summarized in Table 2 and Table 3, respectively.

In cycle A, single or double bursts forming bundles in the reticulum, and a single bundle in the ventral sac of rumen were described using amplitude, RMS, and duration parameters. There were no differences in amplitude between the first and the second burst in the reticulum (1.93 ± 0.29 versus 1.74 ± 0.30 mV, *p* = 0.511), whereas for both the amplitude was always higher (*p* < 0.0001) than in the rumen (0.72 ± 0.14 mV). The coefficient of variation for amplitude was higher in the rumen than in the reticulum across animals (22.16% vs. 11.13–14.30%) and throughout observations (20.38% vs. 13.19–16.87%). The RMS of the second burst in the reticulum was higher than in the first (0.48 ± 0.06 versus 0.35 ± 0.05 mV, *p* = 0.0041), and again for both the RMS was always higher (*p* < 0.0001) than in the rumen (0.10 ± 0.02 mV). The coefficient of variation for RMS was lower in the rumen than in the reticulum across animals (19.29% vs. 26.05–47.62%) and throughout observations (20.38% vs. 24.57–45.69%). On the other hand, the burst lasted longer (*p* < 0.0001) in the rumen (7.71 ± 1.13 s) in comparison to the reticulum, where the first burst lasted significantly longer than the second (1.95 ± 0.26 versus 0.82 ± 0.49 s, *p* < 0.0001). The coefficient of variation for the duration was similar in the rumen and reticulum across animals (18.53% vs. 12.26–18.96%) and throughout observations (14.96% vs. 11.86–13.74%).

In cycle B, single or double bursts forming bundles in the reticulum, and double bursts forming bundles in the ventral sac of rumen were similarly described. There were no differences (*p* > 0.05) in amplitude and RMS between the first (A: 0.75 ± 0.15 mV; RMS: 0.11 ± 0.02 mV) and the second (A: 0.71 ± 0.19 mV; RMS: 0.13 ± 0.02) burst in the rumen, however, both parameters were always lower (*p* < 0.0001) than in the reticulum. The RMS of the second burst in the reticulum was higher than of the first (0.37 ± 0.05 versus 0.51 ± 0.07 mV, *p* = 0.002) with no differences in amplitude (1.89 ± 0.31 versus 1.73 ± 0.36 mV, *p* = 0.149). The coefficient of variation for amplitude was similar in the rumen and in the reticulum across animals (14.34–20.16 vs. 19.66–24.55%) and throughout observations (18.74–26.53% vs. 15.33–20.66%). The coefficient of variation for RMS was lower in the rumen than in the reticulum across animals (10.44–15.09% vs. 27.60–51.57%) and throughout observations (11.24–16.33% vs. 24.77–50.54%). The duration of the first burst lasted always longer than in the second burst, in both the rumen (7.85 ± 1.23 versus 5.27 ± 0.75 s, *p* < 0.01) and the reticulum (2.01 ± 0.19 versus 0.73 ± 0.12 s, *p* < 0.01). Again, the burst lasted longer (*p* < 0.0001) in the rumen than in the reticulum. The coefficient of variation for the duration was similar in the rumen in reticulum across animals (13.34–17.54% vs. 13.40–16.51%) and throughout observations (10.91–15.66% vs. 15.62–19.36%).

In cycle C, triple bursts forming bundles in the reticulum, and single or double bursts forming bundles in the ventral sac of rumen were described, likewise. The burst parameters in the rumen were similar to cycle B, with no differences in amplitude and RMS, and longer duration of the first burst. Additionally, the burst parameters in the reticulum corresponded to those in cycle B in the case of the first and the second burst. However, the third burst (0.78 ± 0.08 s) was shorter (*p* < 0.0001) than the first burst (2.77 ± 0.14), with no differences (*p* = 0.113) with the second one (1.06 ± 0.27 s). The coefficient of variation for the duration was higher in the rumen than in the reticulum across animals (12.80–52.69% vs. 12.47–18.26%) and throughout observations (13.70–53.22% vs. 13.09–17.17%). There were also no differences in amplitude (*p* = 0.263) and RMS (*p* = 0.132) between the third burst (A: 1.60 ± 0.28 mV; RMS: 0.45 ± 0.16 mV) and the previous two in the reticulum. The coefficient of variation for amplitude was similar in the rumen and reticulum across animals (10.66–29.00% vs. 10.83–20.82%) and throughout observations (19.79–24.11% vs. 15.91–22.86%). The coefficient of variation for RMS was slightly lower in the rumen than in the reticulum across animals (13.68–19.23% vs. 17.73–36.22%) and throughout observations (16.97–17.54% vs. 21.69–35.61%).

Summarizing the values of the general parameters of bundles in all three types of the reticulo-rumen cycles, there were no differences (*p* > 0.05) in amplitude and RMS of bundles recorded both in the rumen and in the reticulum, between cycle A, cycle B, and cycle C. At the bundles level, the major differences between cycles concerned duration.

In the rumen, bundles lasted the longest (*p* < 0.0001) in cycle B (26.14 ± 2.68 s) in comparison to shorter bundles in cycle C (16.79 ± 8.94 s), and the shortest in cycle A (7.71 ± 1.13 s). The observed difference resulted from the recorded number of bursts forming bundles, and the length of pauses between subsequent bursts. In the reticulum, the durations of the bundles were comparable (*p* = 0.095) in cycles A (3.24 ± 0.37 s) and B (3.25 ± 0.22 s), whereas in cycle C (6.48 ± 0.39 s) the bundle lasted longer (*p* < 0.05), due to the presence of an additional, third burst in the reticulum. Generally, the amplitude and RMS of myoelectrical activity were higher in the reticulum than in the rumen, however, were shorter, regardless of the type of the reticulo-rumen cycles.

## 4. Discussion

The analysis of the physiological motor activity of the gastrointestinal tract of ruminants is an important element of both basic and applied research in the context of the clinical application [4,17,39,40,41,42,43]. The latest studies characterizing the motor activity of the forestomachs carried out using ultrasound techniques [44,45] give way to EMG studies of the gastrointestinal tract in sheep [4] and pigs [13,23], which were considered crucial for understanding the mechanisms regulating the motor activity of the gastrointestinal tract and for clinical practice.

The obtained EMG recordings were subjected to a detailed off-line analysis taking into account the assessment of the parameters of bundles and, carried out for the first time, an analysis of the parameters of the bursts forming the bundles. To perform a comprehensive characterization of the physiological myoelectric activity of the smooth muscle of the rumen and the reticulum in sheep, a division of the reticulo-ruminal was proposed. The presented characteristic of cycles A, B, and C connects the available theoretical knowledge [5,15,30] with the novel, detailed pattern of EMG activity obtained during long-term registration in the reticulo-rumen.

The general pattern of EMG signals presented in this paper is comparable to the characteristic of analog records obtained by Ruckebusch [5], Plaza, [4], and Kania et al. [43] in consecutive segments of the gastrointestinal tract. The average frequency of occurrence of the reticulo-rumen cycles at rest according to Ruckebusch and Bardon [5] is about 1/minute (0.8–1.2 cycles/minute), and according to Forbes and Barrio [30] 1–3 cycles/minute. On the other hand, Kania et al. [42] estimated that, physiologically, sheep are having around three reticulo-ruminal cycles within 2 min (1.5 cycles/minute). In our studies, the number of ruminal cycles remained constant 12 ± 0.2 to 16 ± 0.1 in 15 min intervals, which is in the range of 0.8–1.1 cycles/minute recorded at rest with no food stimulation. Therefore, it can be concluded that the detectability of the reticulo-rumen cycles based on myoelectrical activity registration by a telemetry method is comparable with the results obtained with other recording techniques at rest. It can be concluded that the facility and compatibility of obtained data allow their subject a detailed analysis of interrelationships. However, it should be kept in mind, that the pressure of the ingesta inside the reticulo-rumen and the chemical composition of the ingesta are triggers for the reticulo-rumen motility [1,2,3,4]. Therefore, the data presented here reflected only the basic reticulo-rumen motility pattern at rest, and could be the reason for the relatively low number of ruminal cycles observed over time. When the physiological motility without feed restriction was taken into account, the rumination time (total time spent to rumination per day) ranged from 6.53 to 8.21 h depending on the amount of fiber in the feed [3,46]. In the study presented here, at rest with no food stimulation, the rumination time appears quite short as estimated as 4.50 h.

One of the most important parts of this study was to measure the reticulo-rumen myoelectrical activity in conscious, unrestrained sheep. Measuring methods recently used in ruminants, such as force transducer method [11,12], restrained electromyography [4], or ultrasonographic examination [7,8] required an animal restriction. On the other hand, an application of accelerometers in bolus wireless sensors [9,10] does not allow long-term registration. Therefore, since the myoelectric activity of smooth muscle of the reticulo-rumen reflects its contractile activity [1,2,3], telemetry EMG recording appears to be currently the only method that allows measures of long-term gastrointestinal motility on the unrestrained animal. Moreover, the use of the long-term telemetry method eliminates the greatest limitation in experimenting on the in vivo model in the form of registration and signal transmission to the data processing and archiving system. The long-term telemetry method provides continuous recording and real-time radio signal transmission from animals moving freely in their cages, in contrast to the cable method, which requires training of experimental animals and severe immobilization of them during periodic recording [4,22,24].

The comprehensive characteristics of the myoelectric activity of the reticulo-rumen cycle were based on the analysis of the parameters of the EMG signal: duration of periods of activity (T) and inactivity (P), amplitude (A) and root mean square (RMS), not yet used for describing gastrointestinal tract EMG recording. The action potentials described in this way, recorded in the time domain, were considered as burst forming bundles [23,47]. This detailed description made it possible to unambiguously identify closely synchronized sequences of activities described by Ruckebusch and Tomov [15] including mixing (cycle A), gas eructation (cycle B), and rumination (cycle C) [7].

The classification of the reticulo-rumen cycles described in this paper, supports the existing Ruckebusch and Bordon [5], Braun et al. [7,8], and Plaza et al. [4] classification and extends them with the possibility of evaluating individual bursts forming bundles. Braun [7] et al. analyzed the motor activity of a goat’s reticulum and confirmed the occurrence of 1-, 2-, or 3-phase contractions in the goat. However, he could not determine, using only an ultrasound examination, which contraction of the reticulum corresponds to the motor activity of the rumen, and thus which reticulo-ruminal cycle should be classified. The presented studies also confirmed the occurrence of bundles consisting of 1, 2, or 3 bursts in the reticulum, and the methodology used allowed them to be associated with the rumen contractions, and then classifying the bundles registered in the reticulum consisting of 1 and 2 bursts of activity to cycle A or B, while consisting of three bundles of activities up to cycle C. Available literature lacks publications describing in detail the myoelectric activity of bundles in each of the reticulo-rumen cycles. Julia and Latour [48] limited the automatic analysis of the myoelectrical activity of the gastrointestinal tract to the contraction of the reticulum, describing the duration of periods of activity and inactivity for bundles. Only Plaza et al. [4] conducted a more complex computerized analysis of the myoelectrical activity in the gastrointestinal tract of sheep. The authors implanted five triple nickel-chromium electrodes into the muscular membrane of the reticulum, dorsal rumen sac, omasum, abomasum, and duodenum for recording an analog signal using a 10-channel electroencephalograph. In the available literature, no reports are describing all discussed parameters characterizing the burst in each of the bundles, which significantly hinders the discussion. The results obtained for bundles in subsequent cycles A, B, and C correspond with the results obtained by other authors, while the characteristics of burst registered in the reticulum and the rumen were presented for the first time.

Our studies showed the same duration of the bundles registered in the reticulum, in the A and B cycles, and significantly longer in the C cycle. Plaza et al. [4] obtained similar results for the bundles, however, they did not consider bursts’ parameters, due to method limitations. Additionally, Braun et al. [7], in the ultrasound examination, showed a significantly longer duration of the three-phase contraction of the reticulum (in the C cycle) compared to the single and two-phase contraction (in the A and B cycles). We demonstrated also, lower duration of bundles in the reticulum compared to the rumen in each reticulo-ruminal cycle, which corresponds with the result obtained by Plaza et al. [4].

Analysis of signal amplitude and RMS confirmed that those parameters are relatively constant and did not fluctuate significantly inhomogeneous bundles and burst in the reticulum and rumen. In the study of Plaza et al. [4], the amplitude of bundles was averaged over the entire record and was not analyzed for individual bursts. In our study, the amplitude of the bundles in the reticulum was always higher than in the rumen, moreover, the amplitude of the burst forming bundles did not differ between the cycles for the reticulum and rumen, respectively. Similarly, the RMS of the bundles in the reticulum was always higher than in the rumen. The analysis of individual bursts showed an interesting difference between the first and second reticular contractions and the first and second rumen contractions is RMS, but not in amplitude. In recent studies, RMS was not used for the characterization of the motor activity of the forestomach [4,5,16,40]. Plaza et al. [4] describing the second contraction of the reticulum as stronger could not calculate the EMG signal strength in an analog data recording system on paper with a speed of 3–4 cm/min. Only the analysis of the digital signal carried out in the presented work allowed the separation of the amplitude and RMS (as the mean square after the amplitude in the frequency domain) as two dependent, but separately analyzed, parameters. Therefore, it can be concluded that the frequency of electrical events in a burst, as a variable affecting signal strength, may be an important regulator of the reticulum myoelectrical activity, in contrast to the rumen, in which no signal strength differences between burst in cycles B and C were found.

The main limitation of this study is the lack of electrodes placed on the dorsal sac of the reticulum or the atrium. The lack of additional biopotential measurements results from the technical limitation of implantable telemetry for large animals to 3-channel transmitter implant. In the study presented here, three electrodes were positioned on the central section of the cranial part of the reticulum, on the ventral sac of the rumen, and the abomasum. Future research with modified arrangements of electrodes are required to achieve useful information about the contribution of the dorsal sac or the atrium to the reticulo-rumen motility. One of the solutions to this limitation may be an application of multimodal telemetry recordings combining simultaneous measurements of EMG and intraruminal pressure. With the development of technological advances, the availability of specific sensor-based technologies that can monitor these physiological changes increases [49]. Recently, multimodal telemetry was successfully applied to monitoring brain function and intracranial pressure in macaques [50], and may support a new approach in the minimally invasive monitoring of the reticulo-rumen activity in sheep.

## 5. Conclusions

It may be concluded that the long-term EMG telemetry registration is feasible in the assessment of the reticulo-rumen myoelectrical activity in sheep. The obtained results suggest that the EMG signal registered in the wall of the nonstimulated gastrointestinal tract in sheep is not random, but it occurs in an orderly array and percentage. Those EMG signals of each rumination cycle formed a specific pattern of the reticulo-ruminal motility during the rest period. The recognition of the A, B, and C reticulo-rumen cycles ineffective based on the EMG signal structure. The use of bursts parameters allows very detailed characteristics of the reticulo-rumen contractile pattern. These results may be the basis for researching the regulation of gastrointestinal motility using the long-term telemetry method.

## Figures and Tables

**Figure 1 animals-11-01052-f001:**
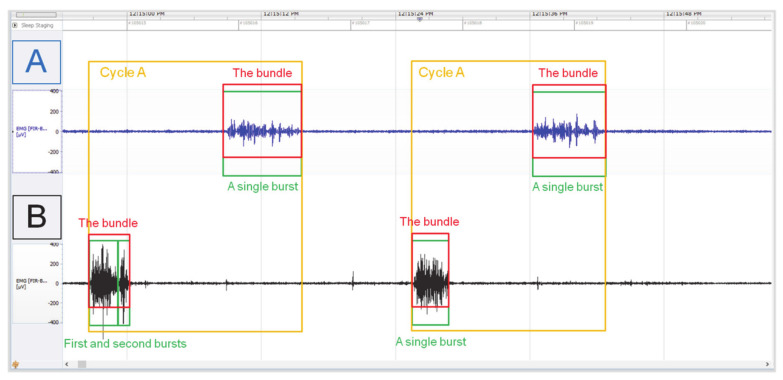
The representative images of the EMG signal recording in the reticulo−rumen, representing cycle A. Myoelectrical activity of the rumen (**A**); myoelectrical activity of the reticulum (**B**). NeuroScore analysis software, 1-min timeline.

**Figure 2 animals-11-01052-f002:**
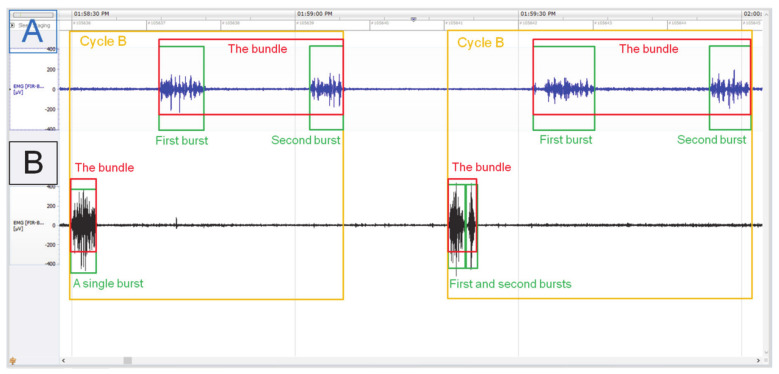
The representative images of the EMG signal recording in the reticulo-rumen, representing cycle B. Myoelectrical activity of the rumen (**A**); myoelectrical activity of the reticulum (**B**). NeuroScore analysis software, 1-min timeline.

**Figure 3 animals-11-01052-f003:**
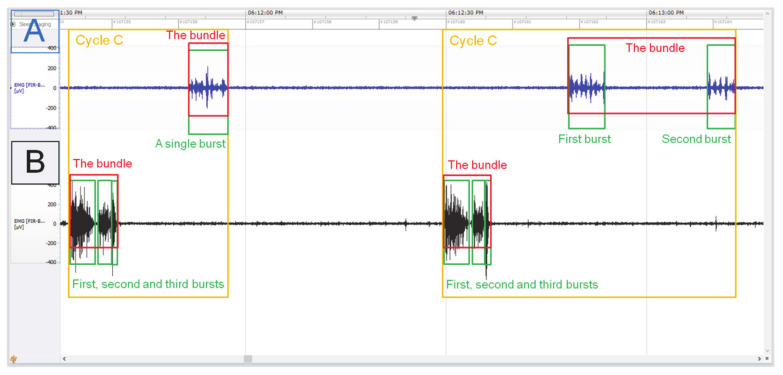
The representative images of the EMG signal recording in the reticulo-rumen, representing cycle C. Selected Myoelectrical activity of the rumen (**A**); myoelectrical activity of the reticulum (**B**). NeuroScore analysis software, 1 min timeline.

**Figure 4 animals-11-01052-f004:**
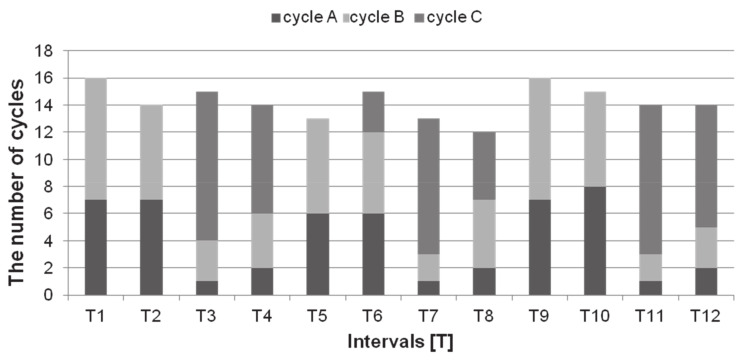
The number of the reticulo-rumen cycles in sheep in the consecutive intervals, including a contribution of cycles A, B, and C.

**Table 1 animals-11-01052-t001:** The electromyography (EMG) signal parameters (mean ± SD) in the reticulo-rumen during cycles A, B, and C registered in sheep (*n* = 9) during 3 h daily recording session.

Reticulo-Rumen	EMG		Cycle A		Cycle B		Cycle C	
	Parameter		Rumen	Reticulum	Rumen	Reticulum	Rumen	Reticulum
	A ^1^	(mV)	0.72 ± 0.14 ^a^	2.01 ± 0.28 ^b^	0.85 ± 0.16 ^a^	1.98 ± 0.30 ^b^	0.80 ± 0.17 ^a^	2.11 ± 0.41 ^b^
Bundle	RMS ^2^	(mV)	0.10 ± 0.02 ^m^	0.51 ± 0.22 ^n^	0.13 ± 0.02 ^m^	0.50 ± 0.001 ^n^	0.12 ± 0.02 ^m^	0.61 ± 0.001 ^n^
	D ^3^	(s)	7.71 ± 1.13 ^u^	3.24 ± 0.37 ^v^	26.14 ± 2.68 ^w^	3.25 ± 0.22 ^v^	16.79 ± 8.94 ^x^	6.48 ± 0.39 ^y^
	A ^1^	(mV)	0.72 ± 0.14 ^a^	1.93 ± 0.29 ^b^	0.75 ± 0.15 ^a^	1.89 ± 0.31 ^b^	0.76 ± 0.18 ^a^	1.76 ± 0.28 ^b,c^
First burst	RMS ^2^	(mV)	0.10 ± 0.02 ^m^	0.35 ± 0.05 ^o^	0.11 ± 0.02 ^m^	0.37 ± 0.05 ^o^	0.11 ± 0.02 ^m^	0.33 ± 0.07 ^o^
	D ^3^	(s)	7.71 ± 1.13 ^u^	1.95 ± 0.26 ^v^	7.85 ± 1.23 ^u^	2.01 ± 0.19 ^v^	7.25 ± 0.99 ^u^	2.77 ± 0.14 ^v^
	A ^1^	(mV)	-	1.74 ± 0.30 ^b,c^	0.71 ± 0.19 ^a^	1.73 ± 0.36 ^b,c^	0.66 ± 0.13 ^a^	1.78 ± 0.41 ^b,c^
Second burst	RMS ^2^	(mV)	-	0.48 ± 0.06 ^n^	0.13 ± 0.02 ^m^	0.51 ± 0.07 ^n^	0.22 ± 0.02 ^m^	0.51 ± 0.17 ^n^
	D ^3^	(s)	-	0.82 ± 0.49 ^z^	5.27 ± 0.75 ^y^	0.73 ± 0.12 ^z^	5.46 ± 0.75 ^y^	1.06 ± 0.27 ^z^
	A ^1^	(mV)	-	-	-	-	-	1.60 ± 0.28 ^c^
Third burst	RMS ^2^	(mV)	-	-	-	-	-	0.45 ± 0.16 ^n,o^
	D ^3^	(s)	-	-	-	-	-	0.78 ± 0.08 ^z^

^1^ A—amplitude, ^2^ RMS—root mean square, ^3^ D—duration of electrical activity. The entries that share a superscript do not differ statistically (*p* > 0.05) between the types of cycles (cycle A/cycle B/cycle C) and place of registration (the rumen/the reticulum) independently for each parameter: A (a, b, c), RMS (m, n, o), and D (v, w, x, y, z).

**Table 2 animals-11-01052-t002:** The coefficient of variation (CV, %) for EMG signal parameters in the reticulo-rumen during cycles A, B, and C calculated across animals (*n* = 9).

Reticulo-Rumen	EMG		Cycle A		Cycle B		Cycle C	
	Parameter		Rumen	Reticulum	Rumen	Reticulum	Rumen	Reticulum
	A ^1^	(mV)	22.16%	11.13%	14.34%	20.55%	26.15%	15.49%
Bundle	RMS ^2^	(mV)	19.29%	44.88%	10.44%	46.30%	19.23%	21.93%
	D ^3^	(s)	18.53%	17.14%	16.21%	13.40%	52.69%	16.08%
	A ^1^	(mV)	22.16%	12.11%	14.74%	19.66%	29.00%	10.83%
First burst	RMS ^2^	(mV)	19.29%	26.05%	11.74%	27.60%	17.94%	17.73%
	D ^3^	(s)	18.53%	18.96%	17.54%	16.51%	12.80%	14.27%
	A ^1^	(mV)	-	14.30%	20.16%	24.55%	10.66%	20.82%
Second burst	RMS ^2^	(mV)	-	47.62%	15.09%	51.57%	13.68%	27.86%
	D ^3^	(s)	-	12.26%	13.34%	13.66%	16.69%	18.24%
	A ^1^	(mV)	-		-	-	-	18.85%
Third burst	RMS ^2^	(mV)	-	-	-	-	-	36.22%
	D ^3^	(s)	-	-	-	-	-	12.47%

^1^ A—amplitude, ^2^ RMS—root mean square, ^3^ D—duration of electrical activity.

**Table 3 animals-11-01052-t003:** The coefficient of variation (CV, %) for EMG signal parameters in the reticulo-rumen during cycles A, B, and C calculated throughout observations (5–6 weeks).

Reticulo-Rumen	EMG		Cycle A		Cycle B		Cycle C	
	Parameter		Rumen	Reticulum	Rumen	Reticulum	Rumen	Reticulum
	A ^1^	(mV)	20.38%	13.19%	18.74%	15.33%	21.12%	16.07%
Bundle	RMS ^2^	(mV)	18.57%	42.85%	11.24%	39.83%	17.54%	21.69%
	D ^3^	(s)	14.96%	11.86%	10.91%	16.82%	53.22%	16.46%
	A ^1^	(mV)	20.38%	15.08%	18.82%	16.18%	24.11%	15.91%
First burst	RMS ^2^	(mV)	18.57%	24.57%	13.94%	24.77%	16.97%	23.25%
	D ^3^	(s)	14.96%	13.23%	15.66%	19.36%	13.71%	15.05%
	A ^1^	(mV)	-	16.87%	26.53%	20.66%	19.79%	22.86%
Second burst	RMS ^2^	(mV)	-	45.69%	16.33%	50.54%	17.21%	33.84%
	D ^3^	(s)	-	13.74%	14.22%	15.62%	13.70%	17.17%
	A ^1^	(mV)	-	-	-	-	-	17.47%
Third burst	RMS ^2^	(mV)	-	-	-	-	-	35.61%
	D ^3^	(s)	-	-	-	-	-	13.09%

^1^ A—amplitude, ^2^ RMS—root mean square, ^3^ D—duration of electrical activity.

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
