# Peer review of "Long-Term Recording of Reticulo-Rumen Myoelectrical Activity in Sheep by a Telemetry Method"

_animals, 2021, doi:10.3390/ani11041052_

Round 1

Reviewer 1 Report

The authors used EMG to monitor the motility of the reticulorumen, and provide data regarding sequence and duration of specific motility patterns based on EM-signals. The manuscript fits the scope of the journal.

Nevertheless, there are some concerns:

Only two sensors were placed at the reticulorumen. What about the dorsal sac or the atrium? It is well known that these structures vitally contribute to reticulorumen motility and EMG in this regions may have added useful information.

What was the reason for the feed limitation (line 148-150)? Since the pressure of the ingesta inside the reticulorumen as well as the chemical composition of the ingesta (pH, SCFA) are triggers of reticulorumen motility, this limitation may lead to a (non-physiological) reduction of motility at all. This casts the results into doubt. It might also be the reason for the relatively low number of ruminal cycles observed over time. Also, the rumination time appears quite short (4.5 hours/d, extrapolated, based on the data provided by the authors). It would highly improve the study to present whole-day recordings, and to compare data before and after feeding.

To evaluate the data and to allow a reasonable interpretation, contractions should be monitored by accelaration or pressure sensors, and compared to EMG data (though, this would hardly be possible by telemetry).

minor points:

line 58-66: Every textbook of veterinary physiology displays the phases of the reticulorumen motility. I do not understand what EMG should add to current knowledge in this respect. As stated above, a comparison between EMG signals and contraction cycles would be necessary to draw the respective conclusions.

Line 89-90: The advantages of telemetry in terms of animal welfare and stress reduction may not even compensate the surgery necessary to implant the sensors…

line 105-107: There are other possibilities to evaluate reticulorumen motility in clinics or agriculture. EMG appears to be one of the least practical solutions.

There are some spelling errors / missing words troughout the manuscript. A careful revision ist recommended.

Author Response

Dear Reviewer,

We sincerely thank you for your response and the valuable comments on our manuscript. We are glad to hear that our manuscript fits the scope of the journal. We are truly grateful for your remarks which have allowed us to improve the manuscript and the recommendation our manuscript may be accepted after this major revision.

Comment R.1.1

Only two sensors were placed at the reticulorumen. What about the dorsal sac or the atrium? It is well known that these structures vitally contribute to reticulorumen motility and EMG in this regions may have added useful information.

Answer to R.1.1

Thank you for this consideration. We agree with your opinion, that the myoelectrical activity of dorsal sac may have added useful information. The number of sensors was limited through technological solutions available on the market. DSI offers multiple telemetry implants options for large animals such as canines, non-human primates, swine, sheep, and similar sized animals that measure biopotentials. Since 2020 only L03 implants was available, which allows to measure biopotentials only from three locations. We decided to placed third electrode on the abomasum. Without the electrode on the abomasum, the proper passage of ingesta could not be assessed.

Following your comment, we added this information to the limitation paragraph in the discussion section.

Comment R.1.2

What was the reason for the feed limitation (line 148-150)? Since the pressure of the ingesta inside the reticulorumen as well as the chemical composition of the ingesta (pH, SCFA) are triggers of reticulorumen motility, this limitation may lead to a (non-physiological) reduction of motility at all. This casts the results into doubt. It might also be the reason for the relatively low number of ruminal cycles observed over time. Also, the rumination time appears quite short (4.5 hours/d, extrapolated, based on the data provided by the authors). It would highly improve the study to present whole-day recordings, and to compare data before and after feeding.

Answer to R.1.2

Thank you very much for this recommendation. The whole investigation was designed to describe the impact of feeding and administration of opioid receptors agonists and antagonists on the reticulorumen and abomasum motility in sheep. The first step was to implement the telemetry method to long-term recording of reticulo-rumen myoelectrical activity in sheep. To the best of our knowledge, this is the first paper of its kind. Therefore, the primary aim of this study was to investigate the feasibility of electromyography registered by a telemetry method for the assessment of the reticulo-rumen myoelectrical activity in sheep. In our basic study, we followed the model of reticulo-rumen cycles estimation at rest proposed by Ruckebusch and Bardon [5], Forbes and Barrio [30] and Kania et al. [42]. The number of ruminal cycles achieved in our studies is in the range of 0.8 - 1.1 cycles/minute following the reference values recorded at rest with no food stimulation [5,30,42].

To avoid confusion, the ancillary aims were clarified and the title were revised.

We agree with your opinion, that ingesta inside the reticulorumen affects reticulorumen motility, and food limitation may lead to a reduction of motility. Therefore we added the feed limitation aspect to the discussion section in line with relative low number of cycles observed over time. Both the whole-day recordings as well as the comparative data before and after feeding will be presented in a subsequent paper.

Comment R.1.3

To evaluate the data and to allow a reasonable interpretation, contractions should be monitored by accelaration or pressure sensors, and compared to EMG data (though, this would hardly be possible by telemetry).

Answer to R.1.3

Thank you very much for pointing it out. One of the most important parts of this study was to measure the reticulo-rumen myoelectrical activity in conscious, unrestrained sheep. As you mentioned, it would not be possible by using acceleration or pressure sensors. It is well known that the myoelectric activity of smooth muscle of the rumen and the reticulum reflects its contractile activity, telemetry EMG recording is currently the only method that allows measures of long-term gastrointestinal motility on the unrestrained animal.

Moreover, the use of the telemetry method eliminates the greatest limitation in experimenting on the in vivo model in the form of registration and signal transmission to the data processing and archiving system. The telemetry method provides continuous recording and real-time radio signal transmission from animals moving freely in their cages, in contrast to the cable method, which requires training of experimental animals and severe immobilization of them during periodic recording

Following your comment, we added this information to the discussion section.

Comment R.1.4

line 58-66: Every textbook of veterinary physiology displays the phases of the reticulorumen motility. I do not understand what EMG should add to current knowledge in this respect. As stated above, a comparison between EMG signals and contraction cycles would be necessary to draw the respective conclusions.

Answer to R.1.4

Thank you for pointing it out. As stated above, telemetry EMG recording is currently the only method that allows measures of long-term gastrointestinal motility on the unrestrained animal. In this preliminary study, we shown recording reticulo-rumen EMG signal with telemetry approach offers a reliable tool to assess the forestomach motility in conscious, unrestricted sheep allows for easy recognition of reticulo-rumen cycles. This stage is required for further long-term investigations on changes of reticulo-rumen cycles in response to a different diet or pharmaceutical treatment.

Following your comment, we added this information to the introduction section.

Comment R.1.5

Line 89-90: The advantages of telemetry in terms of animal welfare and stress reduction may not even compensate the surgery necessary to implant the sensors…

Answer to R.1.5

Thank you for pointing it out. We agree with your opinion and therefore add this information to the introduction section.

 Comment R.1.6

line 105-107: There are other possibilities to evaluate reticulorumen motility in clinics or agriculture. EMG appears to be one of the least practical solutions.

Answer to R.1.6

Thank you for pointing it out. We agree with your opinion and therefore add this information to the introduction section.

Comment R.1.7

There are some spelling errors / missing words troughout the manuscript. A careful revision ist recommended.

 Answer to R.1.7

Thank you very much for this recommendation. The manuscript has been carefully revised and the detected errors were corrected.

Reviewer 2 Report

This is an interesting paper that uses telemetry to record rumen electromyographic signals from sheep. The use of telemetry to record electrical activity in the rumen is a new contribution and offers advantages in the ability to carry out longer term recordings while allowing greater animal motility. Notwithstanding this contribution, the paper cannot be accepted in its current form because of deficiencies in the analysis, presentation and interpretation of the measurements. 

Major concerns:

1) What is the justification for classifying signals as mixing, eructation and rumination patterns based on single, double and triple bursts? No independent (I.e. non-electromyographic) measurements are presented to validate the classification - for example, were visual or observations used to verify that animals were indeed ruminating when sequence C signals were recorded? 

2) Related to the above, the authors state (lines 294-296) that the classification system used here differs from a number of previous studies and that it reduces incorrect identification. Evidence should be presented to support this assertion.

3) Animals were not provided access to food for 12 hours prior to recording. It seems surprising that rumination was observed despite this.

4) The authors should provide an analysis of variability across animals and variability over the 5-6 week period of observations. 

5) Telemetry recordings of rumen motility (but not electromyography) have been reported before, e.g. American Journal of Veterinary Research 47(8):1817-25 (1986), ref 8-11 in Regulatory Peptides, 45, pp371-377  (1993). The authors should do a careful survey of existing studies that have used telemetry to record measurements of rumen motility.

Minor concerns:

1) What is the definition of a bundle and a burst? What criteria were used to split a signal into first and second bursts (e.g. in Fig 1B)?

2) Lines 199, 203: what does “for at least a quarter” and “head reaction” mean?

3) Table 1: The superscripts do not indicate statistical significance; rather entries that share a superscript are not statistically different from each other. The sample size should be indicated. Also, related to major concern 4, how repeatable were these metrics between different animals and over the time period of the study?

4) Recordings were also made from the abomasum according to the methods. Why is that data not presented. 

5)What are the authors' thoughts on multimodal telemetry recordings, e.g. combining EMG with intraruminal pressure measurements? This could be nice addition to the discussion.

Author Response

Dear Reviewer,

Thank you very much for your comments and a substantial amount of time invested in looking over the manuscript. We are grateful for your opinion that our manuscript is an interesting paper and for the opportunity to re-submit the manuscript for consideration in the Animals. The providential critiques made are very welcome and certainly improved the quality of the manuscript. We have addressed all considerations or provided an explanation on a few replies. If there is any remaining concern, we are happy to revisit any point that review considered necessary.

Comment R.2.1

What is the justification for classifying signals as mixing, eructation and rumination patterns based on single, double and triple bursts? No independent (I.e. non-electromyographic) measurements are presented to validate the classification - for example, were visual or observations used to verify that animals were indeed ruminating when sequence C signals were recorded?

Answer to R.2.1

Thank you very much for this question. The classification of signals as mixing, eructation, and rumination patterns was proposed based on Plaza et al. [4] and Braun et al. [7,8] works. Braun et al. [7,8] applied the ultrasonographic examination to describe A and B cycles, as a primary and secondary rumen contraction. Also, Plaza et al. [4] used the terms primary and secondary contractions to describe mixing and eructation in reticulo-rumen using electromyography.

Following Braun et al. [8] there are primary and secondary contraction cycles; the former serves primarily to mix the ingesta, support optimal microbial fermentation, and transport ingesta into the omasum (Constable et al., 1990a), whereas the latter are involved in eructation of gas (Constable et al., 1990b).

Each primary contraction cycle (edit: mixing) starts with biphasic reticular contraction, which is followed by contraction of the anterior blind sac of the rumen and then contraction of the dorsal sac of the rumen in a cranio-caudal direction.

Secondary contraction cycles (edit: eructation of gas) do not involve the reticulum and ruminal atrium and consist of contraction of the dorsal sac and then the ventral sac of the rumen. Therefore, in the study presented here, two consecutive contractions of rumen were observed.

Braun et al. [7,8] used ultrasonographic characteristics of the reticulum and rumen to determine the sequence of contractions in goats and cows. The authors also did not use independent measurements (observations) as reference measurements.

Plaza et al. [4] used fifteen sheep and direct bipolar myoelectric activity measurement. The authors did not use independent measurements, for example, behavioral observations, as reference measurements. Plaza et al. [4] described during rumination, the reticular biphasic spike burst became triphasic. It was the first borderline criterion for the identification of the C cycle.

Braun et al. [7,8]  confirm Plaza's et al. [4] findings that a triphasic contraction of the reticulum occurs when a biphasic contraction is immediately preceded by a so-called rejection contraction, which functions to transport the cud into the esophagus, and it represented rumination. It was the second borderline criterion for the identification of the C cycle in our work.

For these reasons in the study presented here, whenever the reticulum triple contraction occurs before the rumen contraction, the cycle was recognized as rumination (the cycle C). Whenever the reticulum monophasic or biphasic contraction occurs before the single rumen contraction, the cycle was recognized as mixing (cycle A). Finally, whenever the reticulum monophasic or biphasic contraction occurs before the double rumen contraction, the cycle was recognized as an eructation of gas (the cycle B).

           Unfortunately, it is not possible to determine the type of contraction of the forestomachs by external observation of the animal (excluding the palpation through a fistula in the left paralumbar fossa which is not contactless). It would be possible by using acceleration or pressure sensors, however, in this way this would hardly be possible by telemetry. We would like to emphasize, that one of the most important parts of this study was to long-term measure the reticulo-rumen myoelectrical activity in conscious, unrestrained sheep. Therefore, following your comment, we added the subsection - Electromyography recognition of A, B, and C cycles to the Materials and Methods section.

Comment R.2.2

Related to the above, the authors state (lines 294-296) that the classification system used here differs from a number of previous studies and that it reduces incorrect identification. Evidence should be presented to support this assertion.

Answer to R.2.2

Thank you for pointing it out. We agree, this conclusion is exaggerated. The sentence has been corrected.

Comment R.2.3

Animals were not provided access to food for 12 hours prior to recording. It seems surprising that rumination was observed despite this.

Answer to R.2.3

Thank you very much for this recommendation. The whole investigation was designed to describe the impact of feeding and administration of opioid receptors agonists and antagonists on the reticulorumen and abomasum motility in sheep. The first step was to implement the telemetry method to long-term recording of reticulo-rumen myoelectrical activity in sheep. To the best of our knowledge, this is the first paper of its kind. Therefore, the primary aim of this study was to investigate the feasibility of electromyography registered by a telemetry method for the assessment of the reticulo-rumen myoelectrical activity in sheep. In our basic study, we followed the model of reticulo-rumen cycles estimation at rest proposed by Ruckebusch and Bardon [5], Forbes and Barrio [30] and Kania et al. [42]. All those authors described reticulo-rumen activity in the period without feeding. Also, Plaza et al. [4] fed sheep with pelleted lucerne hay twice a day at 09.00 and 17.00, and they have received and evaluated rumination in control group without feeding.

We agree with your opinion, that it should be clarified. Therefore we added the feed limitation aspect to the discussion section in line with relative low number of cycles observed over time.

Comment R.2.4

The authors should provide an analysis of variability across animals and variability over the 5-6 week period of observations.

Answer to R.2.4

Thank you for pointing out this deficiency. The coefficient of variation for EMG signal parameters in the reticulo-rumen during cycles A, B and C calculated across animals and over the period of observations have been added.

Comment R.2.5

Telemetry recordings of rumen motility (but not electromyography) have been reported before, e.g. American Journal of Veterinary Research 47(8):1817-25 (1986), ref 8-11 in Regulatory Peptides, 45, pp371-377  (1993). The authors should do a careful survey of existing studies that have used telemetry to record measurements of rumen motility.

Answer to R.2.5

Thank you very much for this recommendation. In the introduction section, we introduced the bolus-type wireless sensor as an available telemetry methods to transmit recordings of rumen motility [9,10,12]. It seems to us that the citation of modern works from 2019, here is more appropriate than the first reports on radio signal transmission from 1970-1981. Therefore, following your comment, we specified the recent use of telemetry to record measurements of rumen motility. 

Comment R.2.6

What is the definition of a bundle and a burst? What criteria were used to split a signal into first and second bursts (e.g. in Fig 1B)?

Answer to R.2.6

Thank you for pointing out this deficiency. The definitions of burst and bundles has been added.

Comment R.2.7

Lines 199, 203: what does “for at least a quarter” and “head reaction” mean?

Answer to R.2.7

Thank you for pointing it out. Sentences have been corrected.

Comment R.2.8

Table 1: The superscripts do not indicate statistical significance; rather entries that share a superscript are not statistically different from each other. The sample size should be indicated. Also, related to major concern 4, how repeatable were these metrics between different animals and over the time period of the study?

Answer to R.2.8

Thank you very much for this recommendation. The missing data has been suplemented.

Comment R.2.9

Recordings were also made from the abomasum according to the methods. Why is that data not presented.

Answer to R.2.9

Many thanks for this question. The third electrode on the abomasum was used to follow the proper passage of ingesta. In sheep, the abomasum contraction does not affect reticulo-rumen motility which was the major aim of this study. To avoid confusion, this information has been added.

Comment R.2.10

What are the authors' thoughts on multimodal telemetry recordings, e.g. combining EMG with intraruminal pressure measurements? This could be nice addition to the discussion.

Answer to R.2.10

Many thanks for this important advice. The paragraph on new possibilities of biological signals from forestomachs was added as the prospect of further research and counterweight to the limitations of this study.

Reviewer 3 Report

The authors must brush up the text in terms of English language.

For example line 67-69....

Author Response

Comment R.3.1

The authors must brush up the text in terms of English language. For example line 67-69....

Answer to R.3.1

Thank you very much for this recommendation. The manuscript has been carefully revised and the detected errors were corrected.

Round 2

Reviewer 1 Report

The provided revision definitely improves the paper. The authors now clearly state that this is a preliminary study and they discuss the limitations of their work. They also provide additional information regarding methods and results that help the reader to follow the statements.

One minor point remaining:

Please, check spelling! Especially in the recently inserted paragraphs, there are wrong/missing words or letters.

Author Response

Dear Reviewer,

We sincerely thank you for your consideration. We are very pleased that provided revision definitely improves the paper.

Comment R.1.1

Please, check spelling! Especially in the recently inserted paragraphs, there are wrong/missing words or letters.

Answer to R.1.1

Thank you very much for this recommendation. We sent the manuscript is a native English external editor. We hope, the revised manuscript has been properly corrected. We hope that the remaining language errors will be corrected as we proceed in the Animals Journal. An important advantage of Animals Journal is providing language proofreading after the article is accepted for publication.

Reviewer 2 Report

The authors have done a commendable job of addressing the issues raised in my previous report. The paper can be accepted for publication, subject to some minor revisions listed below.

1) The previous title was a better description of the primary aim of the study. I suggest retaining it.

2) Line 51: "EMG signal is transferred" - EMG is a measurement technique; what is transferred in the animal is the myoelectrical signal.

3) Line 73: "to recognition" - the word "to" should be removed. Other similar, small grammatical errors should be corrected.

4) Line 148: It is not clear what is meant by "proper passage of ingesta" or why it is necessary to determine this.

5) Figure 4: It is not clear what is being compared to determine statistical significance. Each interval is said to have a significant difference, but from what? For example, in intervals T1 and T2, does it mean that the number of C cycles is statistically different from A and B cycles, while in T8 does it mean that A cycles are different from B and C cycles? And what is the relevance of difference with each interval, since 30 minute intervals are chosen arbitrarily? Would it be better to use this figure as a summary of observations and remove the statistical analysis? The corresponding text (lines 237 - 245) should refer to the intervals when the observations were made, e.g. Cycles C observed in T3-4, T6-8, T11-12 (etc)

Author Response

Dear Reviewer,

We sincerely thank you for your consideration. We are very pleased that provided revision definitely improves the paper.

Comment R.1.1

Please, check spelling! Especially in the recently inserted paragraphs, there are wrong/missing words or letters.

Answer to R.1.1

Thank you very much for this recommendation. We sent the manuscript is a native English external editor. We hope, the revised manuscript has been properly corrected. We hope that the remaining language errors will be corrected as we proceed in the Animals Journal. An important advantage of Animals Journal is providing language proofreading after the article is accepted for publication.

Reviewer 2

Dear Reviewer,

Thank you very much for your consideration. We are grateful for your opinion that our manuscript can be accepted for publication. We are very pleased with kindly words on our revision. We have addressed all minor comments listed below.

Comment R.2.1

The previous title was a better description of the primary aim of the study. I suggest retaining it.

Answer to R.2.1

Thank you for this recommendation. The previous title has been retained.

Comment R.2.2

Line 51: "EMG signal is transferred" - EMG is a measurement technique; what is transferred in the animal is the myoelectrical signal.

Answer to R.2.2

Thank you for pointing it out. The error has been corrected.

Comment R.2.3

Line 73: "to recognition" - the word "to" should be removed. Other similar, small grammatical errors should be corrected.

Answer to R.2.3

Thank you for pointing it out. The error has been corrected. We sent the manuscript is a native English external editor. We hope, the revised manuscript has been properly corrected. We hope that the remaining language errors will be corrected as we proceed in the Animals Journal. An important advantage of Animals Journal is providing language proofreading after the article is accepted for publication.

Comment R.2.4

Line 148: It is not clear what is meant by "proper passage of ingesta" or why it is necessary to determine this.

Answer to R.2.4

Thank you very much for this recommendation. To avoid confusion, this sentence has been rewritten.

Comment R.2.5

Figure 4: It is not clear what is being compared to determine statistical significance. Each interval is said to have a significant difference, but from what? For example, in intervals T1 and T2, does it mean that the number of C cycles is statistically different from A and B cycles, while in T8 does it mean that A cycles are different from B and C cycles? And what is the relevance of difference with each interval, since 30 minute intervals are chosen arbitrarily? Would it be better to use this figure as a summary of observations and remove the statistical analysis? The corresponding text (lines 237 - 245) should refer to the intervals when the observations were made, e.g. Cycles C observed in T3-4, T6-8, T11-12 (etc)

Answer to R.2.5

Thank you very much for pointing this insufficiency out. We agree with your opinion. The figure 4 with description and the corresponding paragraph have been corrected.